# Trace Mineral Source Impacts Volatile Fatty Acid Profile and Rumen Trace Mineral Solubility in Feedlot Steers [note 1]

**DOI:** 10.3390/ani15091271

**Published:** 2025-04-30

**Authors:** Huey Yi Loh, Octavio Guimaraes, Meghan P. Thorndyke, Sam Jalali, Jerry W. Spears, Jeff S. Heldt, Terry E. Engle

**Affiliations:** 1Department of Animal Sciences, Colorado State University, Fort Collins, CO 80523, USA; huey.loh@colostate.edu; 2Midwest PMS LLC, Englewood, CO 80112, USA; oguimaraes@mwpms.com (O.G.); mthorndyke@mwpms.com (M.P.T.); 3Pilgrim’s Pride Company, Greeley, CO 80634, USA; sam.jalali@pilgrims.com; 4Department of Animal Sciences, North Carolina State University, Raleigh, NC 27695, USA; jspears@ncsu.edu; 5Selko^®^ USA, Indianapolis, IN 46231, USA; jeff.heldt@selko.com

**Keywords:** pH, dialysis, bioavailability

## Abstract

Trace mineral (TM) supplementation in feedlot diets is a common practice in the cattle industry to prevent deficiencies and to improve animal performance. Utilizing different TM sources can alter rumen fermentation due to TM solubility in the rumen. This study supplemented TMs (Cu, Mn, and Zn) from two different sources, either sulfate (STM) or hydroxychloride (HTM), to steers consuming a high concentrate feedlot diet. No differences in nutrient digestibility were observed between TM treatments. However, we observed lower molar proportions of propionate, greater ruminal solubility of Cu, and greater binding strength of Cu and Zn toward digesta in STM-supplemented steers than HTM-supplemented steers. This study suggests that TM source impacts rumen fermentation characteristics and TM solubility in the rumen. This study also suggests that HTM supplementation can result in greater energy production and muscle deposition in animals compared to STM, which can lead to higher yield in feedlot cattle.

## 1. Introduction

Copper, Mn, and Zn are typically supplemented to finishing steer diets to prevent deficiencies due to the inconsistency in concentrations and bioavailability of Cu, Mn, and Zn in the basal dietary ingredients [1]. Sulfate forms of trace mineral (STM) are the most commonly used sources in diet formulation because they are cheaper and more available on the market than other trace mineral (TM) sources [2]. Hydroxychloride TM sources of Cu, Mn, and Zn (HTM) are relatively new sources of TMs in the feed industry. The major difference between STM and HTM is their solubility in the rumen environment. Sulfate TMs are highly soluble in the rumen environment due to weak ionic bonds between the metal and sulfate molecule, whereas HTM are insoluble in the rumen environment because of their covalent bonding structure [2,3,4,5,6]. Despite having differences in ruminal solubility, both STM and HTM have similar solubility at low pH environments [7,8]. This suggests that both sources of TMs will be equally soluble post gastric digestion to allow for absorption in the small intestine.

Hydroxychloride sources of Cu, Mn, and Zn supplementation have been reported to improve fiber digestion [4,5,9,10] and rumen total VFA concentration [4,5] compared to STM sources of Cu, Mn, and Zn supplementation in steers or cows consuming medium-quality forage or dairy-type diets (neutral detergent fiber (NDF) range: 25.3–49.0%). In addition, Spears et al. [11] observed improvements in carcass characteristics in feedlot steers fed a high-concentrate diet supplemented with 18 mg Cu, 40 mg Mn, and 90 mg Zn/kg dry matter (DM) from HTM for 154 d compared to STM-supplemented steers. However, total VFA concentrations were similar across treatments (HTM vs. STM) in that study [11]. These data are in contrast to other studies that reported increases in molar proportions of VFA and total VFA concentrations in HTM- compared to STM-supplemented steers consuming high fiber diets [4,5]. Limited studies have investigated the impact of TM source on fermentation parameters in steers consuming a high concentrate feedlot diet. Therefore, the objectives of the present studies were to determine the influence of TM source on (1) nutrient digestibility and ruminal fermentation parameters, (2) soluble concentrations of Cu, Zn, and Mn in the rumen following a pulse dose of TMs, and (3) binding strength of Cu, Zn, and Mn from ruminal solid digesta in steers fed a diet formulated to meet the National Academies of Sciences, Engineering, and Medicine (NASEM) [12] requirements of a finishing feedlot steer. We hypothesized that steers supplemented with HTM would have greater nutrient digestibility and ruminal fermentation parameters, lower ruminal-soluble Cu, Mn, and Zn concentrations in the rumen, and lower binding strength of Cu, Mn, and Zn to digesta in the rumen compared to STM-supplemented steers.

## 2. Materials and Methods

### 2.1. Ethical Statement

All experimental procedures described in this study were approved by the Colorado State University Animal Care and Use Committee (IACUC approval #17-7182A) prior to the initiation of the experiments.

### 2.2. Dietary Trace Mineral Supplementation Source

Two sources of Cu, Mn, and Zn were used throughout the experiment: (1) sulfate (STM): Cu sulfate pentahydrate (CuSO_4_·5H_2_O; Prince Corporation, Marshfield, WI, USA), Mn sulfate monohydrate (MnSO_4_·H_2_O; Prince Corporation, Marshfield, WI, USA), and Zn sulfate monohydrate (ZnSO_4_·H_2_O; Prince Corporation, Marshfield, WI, USA), and (2) hydroxychloride (HTM): IntelliBond C (Cu_2_(OH)_3_Cl_2_; Selko USA, Indianapolis, IN, USA), IntelliBond M (Mn_2_(OH)_3_Cl; Selko USA, Indianapolis, IN, USA), and IntelliBond Z (Zn_5_(OH)_8_Cl_2_·H_2_O; Selko USA, Indianapolis, IN, USA).

### 2.3. Experiment 1

#### Animals, Diets, and Sample Collection

Procedures described herein were adapted from procedures described by Guimaraes et al. [4,5], with slight modifications. Twelve crossbred Angus steers fitted with ruminal cannulae, owned by Colorado State University, were ranked by body weight (BW; initial BW = 530.9 ± 22.7 kg) and allocated in two feedlot pens (*n* = 6 steers/pen). Steers were fed a basal diet formulated to meet the requirements of finishing feedlot steers (Table 1) for 21 d [12]. The basal diet was sent to a commercial laboratory (SDK Laboratories, Hutchinson, KS, USA) for analysis prior to the initiation of the experiment, and the basal diet’s chemical composition is presented in Table 1. Following the 21 d adaptation period, steers were blocked by BW and randomly assigned to one of two dietary treatments. Dietary treatments consisted of 18, 40, and 90 mg of supplemental Cu, Mn, and Zn/kg DM, respectively, from either STM or HTM sources (*n* = 6 steers/treatment). Ground corn was used as the carrier for TM supplementation. The analyzed concentrations of Cu, Mn, and Zn in the supplement were 620, 1350, and 3020 mg/kg DM, respectively. Treatments were top-dressed at 2.96% of the diet and mixed thoroughly with the diet by hand immediately after the basal diet was delivered. Total dietary Cu, Mn, and Zn concentrations were 22.1 mg Cu, 64.4 mg Mn, and 114.3 mg Zn/kg DM. Dietary TM supplementation was maintained throughout experiment 1.

After receiving dietary treatments for 7 d, steers remained on their appropriate treatments but were moved indoors and individually housed (2.5 m × 2.5 m pens equipped with automatic waterers, individual feeders, and rubber-matted floors) for 2 d to allow for acclimation to their new environment. Steers were then relocated into individual metabolism stalls (3.0 m × 1.1 m pens equipped with automatic waterers, individual plastic feeders, and rubber-matted floors) for a 5 d acclimation period. During the acclimation period, dry matter intake (DMI) for each steer was determined. At the end of the acclimation period, two steers, one from each treatment, were paired based on their mean DMI for the 5 d collection period (6 pairs total), and each steer within a pair was fed the same amount of feed during the sample-collection period. Feed delivered to each steer within a pair during the collection period was calculated to be 90% of the steer with a lower average DMI within the pair during the acclimation period.

Diets were fed twice daily (60% of the ration in the morning and 40% of the ration in the afternoon) during the 5 d sample-collection period. Feed, fecal, and urine collection was conducted daily throughout the 5 d collection period, as described by Caldera et al. [6]. Briefly, feed samples were collected from the total ration before feed delivery to individual steers. Total fecal output from individual steers was quantified by wet weight, thoroughly mixed, and sampled (10% of total wet weight) daily. Feed and individual fecal samples were stored at −20 °C until further analysis. The urine was collected to maintain the cleanliness of the pen and to prevent contamination of samples, so it was not analyzed and was discarded after collection.

After 5 d of sample collection (on d 6), ruminal samples were collected at 0, 2, and 4 h post-feeding to determine ruminal pH and volatile fatty acid (VFA) concentration. Ruminal pH was determined using a portable pH meter (Hanna Instruments, Smithfield, RI, USA). Ruminal contents (approximately 250 g) were obtained from the geometric center of the rumen after the rumen content was thoroughly mixed by hand. Ruminal contents were centrifuged at 28,000× *g* at 5 °C for 30 min, and a 2.0 mL aliquot of the supernatant was acidified with 25% (*v*/*v*) meta-phosphoric acid and stored at −20 °C until VFA analysis.

### 2.4. Experiment 2

#### 2.4.1. Animals, Sample Collection, and Soluble and Insoluble Trace Mineral Concentrations

At the end of experiment 1, steers remained in their individual pens and treatment and were offered ad libitum access to their basal diet without additional supplemental Cu, Mn, and Zn for 14 days. On d 15, steers received a one-time bolus dose, via the cannula, of their respective TM sources (with ground corn serving as the carrier) to provide 18 mg Cu/kg DM, 40 mg Mn/kg DM, and 90 mg Zn/kg DM. Immediately after bolus dose administration, the rumen contents were thoroughly mixed by hand to distribute the TM supplement across the rumen evenly. Ruminal samples were obtained at 2 h intervals beginning at −4 and ending at 24 h post-dosing, with 0 h being the feeding of the basal diet and administration of bolus. To ensure the collection of representative samples during each collection, ruminal contents were thoroughly mixed by hand before the sample (approximately 250 g) was obtained from the geometric center of the rumen. After each collection time, ruminal samples were centrifuged at 2500× *g*, and the supernatant and pellet were separated and stored at −20 ° C for Cu, Mn, and Zn concentration analysis.

#### 2.4.2. Dialysis of Ruminal Insoluble Digesta

A small portion of the total pellets collected at 0, 12, and 24 h was exposed to dialysis using the procedure described by Caldera et al. [6] and Loh et al. [3], with slight modifications. Briefly, the pellet was dried at 60 °C for 48 h in a forced-air drying oven and ground using a mortar and pestle. The dialysis buffer was made by mixing 0.01 M ethylenediaminetetraacetate (EDTA) in 0.05 M Tris (Tris-EDTA), adjusted to a pH of 6.8. Then, 0.25 g of each sample was mixed with 10 mL of Tris-EDTA buffer and placed into dialysis tubing. The samples within the dialysis tubing were then dialyzed against 1.0 L Tris-EDTA buffer at 4 °C with continuous stirring for 16 h. The buffer was replaced with another liter of fresh Tris-EDTA buffer, and dialysis continued for another 6 h. Samples were removed from dialysis bags, placed into pre-weighed acid-washed crucibles, and dried overnight at 60 °C.

### 2.5. Laboratory Analysis

Fecal and feed samples from experiment 1 were proportionally composited across all collection days for each animal for subsequent DM, NDF, acid detergent fiber (ADF), crude protein (CP), and starch analysis. Samples were first dried in a forced-air drying oven for 48 h at 100 °C to determine DM. Dried samples were then analyzed for NDF and ADF using an Ankom 200 Fiber analyzer (ANKOM Technology, Macedon, NY, USA). Crude protein was quantified using the TruSpec CN Carbon/Nitrogen LECO system (LECO Corp., St Joseph, MI, USA). Starch was determined at a commercial laboratory (SDK Laboratories, Hutchinson, KS, USA) using enzymatic hydrolysis (Technicon Industrial method number SE3-0036FJ4). Digestibility for all nutrients was calculated by using $\frac{Nutrient\;in\;feed-Nutrient\;in\;feces}{Nutrientinfeed}\times 100\%$. Volatile fatty acid concentration and composition were analyzed via Agilent 6890N gas chromatography (Agilent Technologies, Santa Clara, CA, USA).

Mineral analysis was conducted according to procedures described by Loh et al. [3], with slight modifications. Briefly, pellet samples (pre- and post-dialysis) from experiment 2 were dried, weighed, and ashed at 600 °C in a Thermo-Fisher Thermolyne muffle furnace overnight after drying. Ashed samples were re-suspended in 5 mL of boiling 1.2 M hydrochloric acid and analyzed for Cu, Mn, and Zn. One mL of supernatant samples and 10 mL of 70% nitric acid were added to digestion vessels and microwave-digested using a Titan MPS microwave digestor (PerkinElmer, Waltham, MA, USA). All samples were then analyzed for Cu, Mn, and Zn concentrations using NexION inductively coupled plasma mass spectrometry (PerkinElmer, Waltham, MA, USA).

### 2.6. Statistical Analysis

All statistical analysis was conducted using SAS software 9.4 [13]. Total tract apparent digestibility of DM, ADF, NDF, starch, and CP was analyzed using a mixed-effects model (PROC MIXED) for a completely randomized block design. A mixed-effects model with repeated-measures analysis (PROC MIXED) for a completely randomized block design was used to analyze all response variables collected over time. Treatment, time, and the treatment × time interaction were considered as fixed effects. The individual animal was considered as the random variable. For all response variables measured, the individual animal was considered the experimental unit. Several covariance structures were compared to determine the most appropriate covariance structure for data analysis. For all response variables, significance was determined at *p* ≤ 0.05, and tendencies were determined at *p* > 0.05 and ≤0.10. When a significant treatment × time interaction or main effect of treatment or time was detected, means were separated using the PDIFF option of the LSMEANS statement of SAS.

## 3. Results

### 3.1. Experiment 1

The influence of trace mineral source on the digestibility of DM, ADF, NDF, CP, and starch in steers receiving a high-concentrate diet is shown in Table 2. By design, DMI was similar across treatments. There were no treatment effects (*p* > 0.10) for DM, NDF, ADF, CP, or starch digestibility.

Table 3 shows the influence of trace mineral sources on pH, total VFA concentrations, and VFA molar proportions at 0, 2, and 4 h post-dosing in steers receiving a high-concentrate diet. There were no treatment × time interactions for pH, total VFA, or molar proportion of acetic acid, propionic acid, or butyric acid. Ruminal pH decreased (*p* < 0.001) with time after feeding (6.23, 5.60, and 5.27, at 0, 2, and 4 h, respectively). Steers receiving HTM had greater (*p* < 0.05) ruminal molar proportions of propionic acid, but lower (*p* < 0.05) molar proportions of butyric acid compared to STM steers. There was a tendency (*p* < 0.07) for a treatment and time interaction for valeric acid. The molar proportions of valeric acid at 0 h and 4 h were greater than at 2 h in steers consuming HTM. The molar proportion of valeric acid tended to increase over time for STM-supplemented steers.

### 3.2. Experiment 2

The influence of trace mineral source on ruminal-soluble Cu, Mn, and Zn is summarized in Figure 1, Figure 2 and Figure 3, respectively. There was a treatment x time interaction (*p* < 0.002) for ruminal-soluble Cu concentrations. Ruminal-soluble Cu concentrations were greater (*p* < 0.05) at 4, 6, 8, and 16 h post-dosing with STM- compared to HTM-supplemented steers (Figure 1). At 14 h post-dosing, HTM-supplemented steers had more (*p* < 0.05) soluble Cu in the rumen than STM-supplemented steers. There was no treatment or treatment x time interaction for ruminal-soluble Mn or Zn concentrations (Figure 2 and Figure 3, respectively).

Concentrations of Cu, Mn, and Zn in the initial digesta and the percent of minerals released post-dialysis are presented in Table 4. There was a treatment × time interaction (*p* < 0.001) for Cu, Mn, and Zn concentrations in ruminal solid digesta. Copper, Mn, and Zn concentrations in ruminal solid digesta prior to TM dosing were similar (*p* > 0.60) across treatments. At 12 h post-dosing, Cu, Mn, and Zn concentrations in solid digesta were greater (*p* < 0.001) in steers receiving HTM compared with those dosed with STM. Copper and Mn concentrations in digesta were similar (*p* > 0.2) across treatments at 24 h post-dosing. Zinc concentration in ruminal solid digesta of steers dosed with HTM at 24 h post-dosing was greater (*p* < 0.001) than those dosed with STM.

There was a treatment x time interaction (*p* < 0.01) for the percentage of Cu, Mn, and Zn in ruminal solid digesta released during dialysis against Tris-EDTA. Copper, Mn, and Zn released from digesta prior to TM dosing (0 h) were similar (*p* > 0.10) across treatments. At 12 and 24 h post-dosing, the release of Cu and Zn from ruminal solid digesta, after being dialyzed against Tris-EDTA, was greater (*p* < 0.001) in steers receiving a pulse dose of HTM compared to those receiving STM sources of Cu and Zn. Steers that were dosed with STM had a greater (*p* < 0.01) Mn release from solid digesta after dialysis at 12 and 24 h post-dosing compared to HTM-dosed steers.

## 4. Discussion

Dry matter, NDF, ADF, CP, and starch digestibility were not influenced by TM source in the present study. These findings are in contrast to the findings from previous studies using similar experimental designs and procedures [4,5]. Steers consuming a medium-quality grass hay-based diet supplemented with 20, 40, and 60 mg Cu, Mn, and Zn/kg DM, respectively, from HTM, had greater DM, NDF, ADF, and CP digestibility than those receiving STM [4]. When fed a diet formulated for a lactating dairy cow supplemented with 10 mg Cu, 40 mg Mn, and 60 mg Zn/kg DM from either STM or HTM sources, steers receiving STM had lower digestibility of NDF and ADF compared to steers receiving HTM. Other studies [10,14,15] investigating total tract digestibility have also reported improved NDF digestibility in lactating dairy cattle supplemented with HTM sources of Cu, Mn, and Zn. In contrast, in situ NDF digestibility was not impacted by TM source (STM vs. HTM) in steers consuming a lactating dairy cow diet [16], which is similar to the findings of the current study. The diet used in the current study was a high concentrate feedlot finishing diet whereas the previous studies, described above, utilized diets with greater fiber content (NDF range: 25.3–49.0%). Furthermore, the pH of the rumen in the current study averaged 5.72 across both treatments, which is lower than the pH values reported in previous studies [4,5] where diets contained greater forage content (rumen pH range: 6.40–6.64). These data indicate that rumen microorganisms that digest fiber may be more sensitive to soluble Cu, Mn, and Zn concentrations than starch-utilizing microorganisms. In addition, according to Calabrò et al. [17], the pH in this study was lower than the suggested value to guarantee a favorable environment for cellulolytic bacteria.

Total VFA concentrations were not influenced by TM source in this study. Previous studies that utilized similar procedures but where animals were fed diets with higher forage concentrations observed greater VFA production in HTM-supplemented steers than STM-supplemented steers [4,5]. In contrast, another study [14] supplementing 15, 20, and 40 mg/kg of Cu, Mn, and Zn from either STM or HTM to Holstein cows consuming a forage-based diet (NDF: 35%) reported similar total VFA concentrations across treatments. A study conducted by Miller et al. [15] utilizing cannulated Holstein cows receiving two corn silage diets containing different NDF concentrations (32% vs. 36%) observed lower total VFA concentrations in cows supplemented with STM (10, 41, and 64 mg/kg DM of Cu, Mn, and Zn, respectively), receiving the 32% NDF diet compared to HTM-supplemented steers fed iso-amount of Cu, Mn, and Zn. However, total VFA concentrations were similar across TM sources for cows consuming the 36% NDF diet. The variation between studies suggests there is an interaction between TM source and dietary NDF levels on fermentation characteristics. In addition, an in vitro study by Loh et al. [18] utilizing rumen fluid collected from crossbred Angus steers consuming a lactating dairy cow diet supplemented with 10 mg Cu, 40 mg Mn, and 60 mg Zn/kg DM from either STM or HTM showed a 40% increase in total VFA production at 24 h post-incubation when rumen fluid from HTM-supplemented steers was utilized. This may suggest that a longer time may be needed to observe the effect of TM source on total VFA production. However, further research is needed since rumen volume and passage rate were not measured in any of the previously mentioned studies.

The greater ruminal molar proportions of propionic acid in steers receiving HTM observed in this study are consistent with previous studies in feedlot steers consuming high-concentrate diets [11]. Spears et al. [11] reported greater ruminal propionate concentrations on d 28 (of a 154 d experiment) in feedlot cattle fed a high concentrate finishing diet with supplemental 18, 40, and 90 mg of Cu, Mn, and Zn/kg DM, respectively, from HTM compared to STM sources. In contrast, the molar proportions of propionic acid and butyric acid were not influenced by TM source (STM or HTM) in steers fed a dairy-type diet [5]. Although no increase in propionic acid molar proportion was observed in steers supplemented with HTM, a lower molar proportion of butyric acid was observed in cattle fed a high-forage diet [4,14]. Results from the current study and previous studies suggest that TM source can impact rumen fermentation, and HTM supplementation may lead to a shift in the ruminal microflora yielding a great production of propionic acid.

Copper, Mn, and Zn concentrations in the rumen digesta at 12 h post-dosing were lower in STM-dosed steers compared to HTM-dosed steers. This may indicate that more Cu, Mn, and Zn from STM-supplemented steers was released into the rumen environment. However, soluble Mn and Zn were similar across TM sources throughout all time points, and soluble Cu from STM was similar to HTM at 12 h post-dosing. We observed greater soluble Cu concentrations in the rumen of steers dosed with STM compared to steers dosed with HTM at 4, 6, 8, and 16 h post-dosing. These findings follow a similar trend to that reported in previous studies where reduced concentrations of soluble Cu were observed in the rumen of HTM- compared to STM-dosed steers consuming diets consisting of either grass hay [4], 50% corn silage 50% steam-flaked corn [6], or a diet formulated for lactating dairy cows [5]. However, ruminal-soluble Mn and Zn concentrations did not differ among the treatments in the current study. This is in contrast to findings from previous studies [4,5,6], where they observed higher levels of soluble Mn and Zn in STM-dosed steers compared to HTM-dosed steers. The reason for the variable responses between experiments may be due to the impact of diet type on rumen pH, as described previously.

The extent to which trace minerals are tightly bound within these insoluble complexes determines whether they can become soluble in the small intestine and be absorbed [19]. In the present study, the percentage of Cu and Zn released from the solid digesta of steers administered HTM was greater when subjected to dialysis against Tris-EDTA at 12 and 24 h after dosing compared to steers receiving STM. In contrast, Mn in digesta obtained from STM-supplemented steers appeared to be more loosely bound to the digesta compared to Mn from HTM-supplemented steers. The binding strengths of Cu and Zn in HTM-supplemented steers were similar to those in studies that fed corn silage and steam-flaked corn-based diet [6], a medium-quality grass hay diet [4], and a lactating dairy cow diet [5]. The release of Mn from STM-supplemented steers was greater than from HTM-supplemented steers at 12 and 24 h in the current study, which is similar to the result from steers consuming a lactating dairy cow diet [5]. In contrast, studies conducted in steers fed medium-quality grass hay [4] and steers fed corn silage and steam-flaked corn-based diets [6] observed similar percentages of Mn released across treatments. The reason for the different particle-binding responses across studies is still unknown, but it could be due to the difference in rumen pH, microbial population, or the interaction between diet particles and TMs.

## 5. Conclusions

The impact of TM source on nutrient digestibility appears to vary depending on diet type. Previous research has reported that beef and dairy cattle consuming diets higher in fiber supplemented with HTM had improved fiber digestibility and rumen fermentation characteristics, and reduced ruminal Cu and Zn solubility compared to STM-supplemented cattle. In the current study, supplementation of HTM to steers consuming a high-concentrate diet increased ruminal molar proportions of propionic acid, decreased ruminal molar proportions of butyric acid, and decreased ruminal-soluble Cu concentrations following a pulse dose of HTM when compared to STM-supplemented steers. Supplementing HTM instead of STM in feedlot steers can promote more propionic acid production, which can eventually lead to more energy production for greater muscle growth and animal productivity. Following dialysis of ruminal digesta, steers receiving a bolus dose of HTM had greater amounts of Cu and Zn released from digesta compared to STM steers, which could indicate greater availability for absorption in the small intestine. However, these differences did not translate into changes in total ruminal volatile fatty acid concentration or improved DM or starch digestion. Factors such as rumen pH, interactions between feed particles and rumen microbiota, free ion concentrations of metals, and binding strength of TMs to digesta and microorganisms may all have a role in how TM source impacts rumen fermentation. Further research is required to understand the specific effects of different trace minerals and TM sources on ruminal fermentation. In addition, utilizing labeled elements in future research will be valuable to understand the proportion of TMs that will be digested, absorbed, and excreted. The excretion of TMs should be paid more attention to in the future as the excretion footprint is becoming an important issue in sustainability.

## Figures and Tables

**Figure 1 animals-15-01271-f001:**
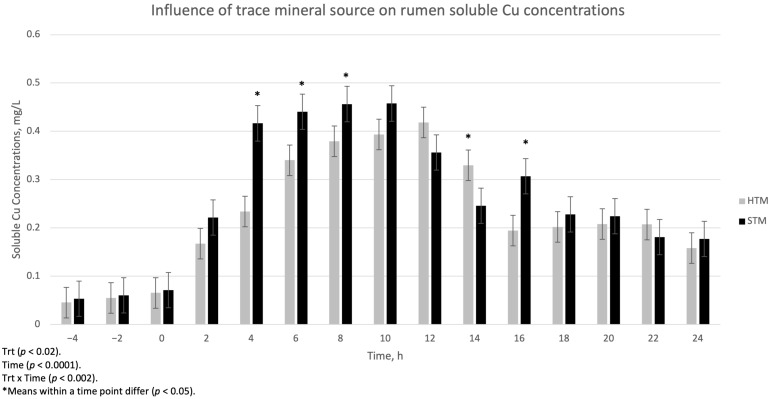
The influence of trace mineral source on soluble Cu within the ruminal contents of steers receiving a pulse dose of either sulfate trace minerals (STM; 18 mg Cu/kg DM from CuSO_4_·5H_2_O; 40 mg Mn/kg DM from MnSO_4_·H_2_O; 90 mg Zn/kg DM from ZnSO_4_·H_2_O) or hydroxy trace minerals (HTM; 18 mg Cu/kg DM from Cu_2_(OH)_3_Cl_2_; 40 mg Mn/kg DM from Mn_2_(OH)_3_Cl; 90 mg Zn/kg DM from Zn_5_(OH)_8_Cl_2_·H_2_O). The *x*-axis denotes sampling time in hours and the *y*-axis denotes ruminal-soluble Cu. Error bars represent standard errors.

**Figure 2 animals-15-01271-f002:**
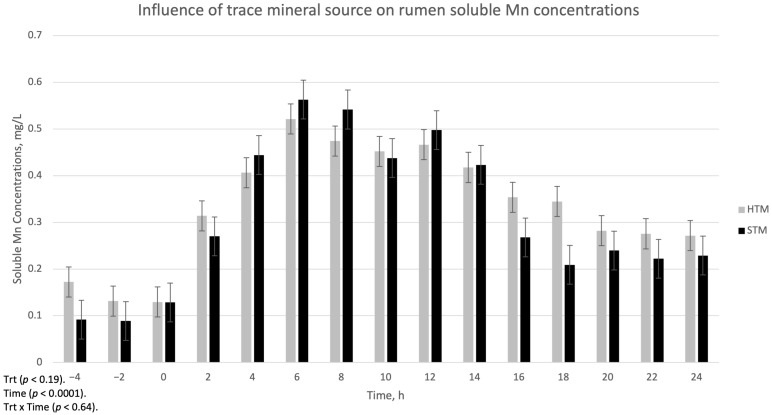
The influence of trace mineral source on soluble Mn within the ruminal contents of steers receiving a pulse dose of either sulfate trace minerals (STM; 18 mg Cu/kg DM from CuSO_4_·5H_2_O; 40 mg Mn/kg DM from MnSO_4_·H_2_O; 90 mg Zn/kg DM from ZnSO_4_·H_2_O) or hydroxy trace minerals (HTM; 18 mg Cu/kg DM from Cu_2_(OH)_3_Cl_2_; 40 mg Mn/kg DM from Mn_2_(OH)_3_Cl; 90 mg Zn/kg DM from Zn_5_(OH)_8_Cl_2_·H_2_O). The x-axis denotes sampling time in hours and the *y*-axis denotes ruminal-soluble Mn. Error bars represent standard errors.

**Figure 3 animals-15-01271-f003:**
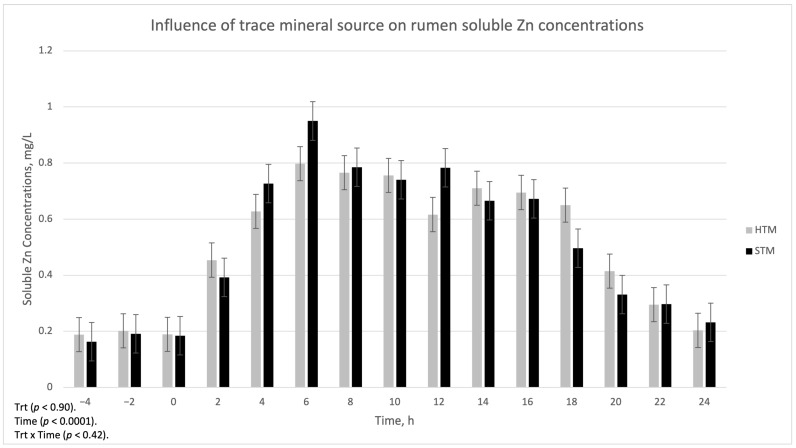
The influence of trace mineral source on soluble Zn within the ruminal contents of steers receiving a pulse dose of either sulfate trace minerals (STM; 18 mg Cu/kg DM from CuSO_4_·5H_2_O; 40 mg Mn/kg DM from MnSO_4_·H_2_O; 90 mg Zn/kg DM from ZnSO_4_·H_2_O) or hydroxy trace minerals (HTM; 18 mg Cu/kg DM from Cu_2_(OH)_3_Cl_2_; 40 mg Mn/kg DM from Mn_2_(OH)_3_Cl; 90 mg Zn/kg DM from Zn_5_(OH)_8_Cl_2_·H_2_O). The *x*-axis denotes sampling time in hours and the *y*-axis denotes ruminal-soluble Zn. Error bars represent standard errors.

**Table 1 animals-15-01271-t001:** Ingredients and analyzed chemical composition of basal diet formulated to meet the requirements of finishing feedlot steers.

Item	
Ingredients, Dry Matter Basis (%)	
Steam-flaked corn	68.9
Corn silage	10.0
Alfalfa hay	9.0
Corn distillers	9.0
Supplement ^1^	3.1
Chemical Composition	
Dry matter, %	65.5
Crude protein, %	13.4
NEm ^2^, Mcal/kg	1.93
NEg ^3^, Mcal/kg	1.41
Fat, %	4.0
Acid detergent fiber, %	10.9
Neutral detergent fiber, %	18.4
Calcium, %	0.71
Phosphorus, %	0.33
Sulfur, %	0.16
Copper, mg/kg	3.74
Manganese, mg/kg	24.43
Zinc, mg/kg	24.8

^1^ Supplement was formulated to provide: 3.5% crude protein equivalents from urea, 0.62% Ca (CaCO_3_), 0.30% salt (NaCl), 0.05% K (KCl), 2401 IU/kg Vitamin A, 15 IU/kg Vitamin E, 33 g/metric ton of monensin (Rumensin 90, Elanco Animal Health, Greenfield, IN, USA), and 11.0 g/metric ton of tylosin (Tylan 100, Elanco Animal Health, Greenfield, IN, USA). ^2^ NEm = net energy for maintenance. ^3^ NEg = net energy for gain.

**Table 2 animals-15-01271-t002:** Influence of trace mineral source on dry matter intake (DMI), dry matter (DM), acid detergent fiber (ADF), neutral detergent fiber (NDF), crude protein (CP), and starch digestibility on steers receiving a high concentrated diet.

Item	Treatment		SEM ^3^	*p*<
	STM ^1^	HTM ^2^		
n=	6	6		
DMI, kg DM/steer/d	8.1	8.1	-	-
DM digestibility, %	75.8	79.5	1.75	0.16
NDF digestibility, %	41.2	41.0	0.40	0.71
ADF digestibility, %	27.2	30.5	1.47	0.15
CP digestibility, %	69.8	68.4	1.13	0.40
Starch digestibility, %	96.7	96.1	0.63	0.52

^1^ STM = 18 mg Cu/kg DM from CuSO_4_·5H_2_O; 40 mg Mn/kg DM from MnSO_4_·H_2_O; 90 mg Zn/kg DM from ZnSO_4_·H_2_O. ^2^ HTM = 18 mg Cu/kg DM from Cu_2_(OH)_3_Cl_2_; 40 mg Mn/kg DM from Mn_2_(OH)_3_Cl; 90 mg Zn/kg DM from Zn_5_(OH)_8_Cl_2_·H_2_O. ^3^ SEM = standard error of the mean.

**Table 3 animals-15-01271-t003:** Influence of trace mineral source on pH and volatile fatty acid production at 0, 2, and 4 h post-feeding in steers receiving a high concentrated diet.

Item	Treatment		SEM ^3^	*p*<		
	STM ^1^	HTM ^2^		Trt	Time	Trt × Time
n=	6	6				
pH						
0 h	6.21	6.25	0.06	0.98	0.0001	0.93
2 h	5.60	5.61				
4 h	5.35	5.32				
Total volatile fatty acids, mM						
0 h	114.60	115.90	2.74	0.80	0.69	0.61
2 h	116.43	122.23				
4 h	120.17	116.20				
Volatile fatty acid, mM/100 mM						
Acetic acid						
0 h	31.95	29.62	2.59	0.64	0.07	0.47
2 h	31.88	31.70				
4 h	30.27	27.35				
Propionic acid						
0 h	30.33	38.50	3.12	0.05	0.0001	0.42
2 h	38.00	47.53				
4 h	38.65	50.98				
Butyric acid						
0 h	16.47	12.12	1.77	0.05	0.53	0.17
2 h	18.18	11.63				
4 h	17.50	11.20				
Valeric acid						
0 h	9.33	11.11	2.11	0.69	0.02	0.07
2 h	9.85	7.25				
4 h	12.33	9.48				

^1^ STM = 18 mg Cu/kg DM from CuSO_4_·5H_2_O; 40 mg Mn/kg DM from MnSO_4_·H_2_O; 90 mg Zn/kg DM from ZnSO_4_·H_2_O. ^2^ HTM = 18 mg Cu/kg DM from Cu_2_(OH)_3_Cl_2_; 40 mg Mn/kg DM from Mn_2_(OH)_3_Cl; 90 mg Zn/kg DM from Zn_5_(OH)_8_Cl_2_·H_2_O. ^3^ SEM = standard error of the mean.

**Table 4 animals-15-01271-t004:** Influence of dialysis on copper, manganese, and zinc release from rumen solid digesta 0, 12, and 24 h after receiving a pulse dose of 18 mg Cu, 40 mg Mn, and 90 mg Zn/kg DM from either hydroxy or sulfate trace mineral sources.

Item	Trace Mineral Source		SEM ^3^	*p*<		
	STM ^1^	HTM ^2^		Trt	Time	Trt × Time
n=	6	6				
Initial concentration of digesta, mg/kg DM						
Cu						
0 h	0.83 ^a^	0.85 ^a^	0.28	0.0001	0.0001	0.0001
12 h	4.7 ^b^	26.6 ^c^	4.01			
24 h	1.2 ^a^	0.66 ^a^	0.45			
Mn						
0 h	7.6 ^a^	7.5 ^a^	0.95	0.0001	0.0001	0.0001
12 h	25.3 ^b^	29.7 ^c^	4.67			
24 h	7.3 ^a^	7.4 ^a^	2.35			
Zn						
0 h	11.8 ^a,b^	10.9 ^a^	0.99	0.0001	0.0001	0.0001
12 h	28.9 ^c^	109.8 ^d^	10.1			
24 h	9.7 ^e^	18.2 ^b^	10.2			
Mineral released post-dialysis, % ^4^						
Cu						
0 h	27.4 ^a,b^	23.6 ^b^	1.27	0.0001	0.0001	0.0001
12 h	27.3 ^a,b^	61.4 ^c^	1.61			
24 h	28.9 ^a^	80.3 ^d^	13.7			
Mn						
0 h	32.2 ^a^	31.9 ^a^	1.93	0.0007	0.0001	0.0025
12 h	79.5 ^b^	64.5 ^c^	16.2			
24 h	98.1 ^d^	90.5 ^e^	10.2			
Zn						
0 h	50.1 ^a^	56.1 ^a^	2.67	0.0001	0.0005	0.0001
12 h	35.3 ^b^	91.7 ^c^	12.4			
24 h	33.3 ^b^	89.8 ^c^	14.9			

^1^ STM = 18 mg Cu/kg DM from CuSO_4_·5H_2_O; 40 mg Mn/kg DM from MnSO_4_·H_2_O; 90 mg Zn/kg DM from ZnSO_4_·H_2_O. ^2^ HTM = 18 mg Cu/kg DM from Cu_2_(OH)_3_Cl_2_; 40 mg Mn/kg DM from Mn_2_(OH)_3_Cl; 90 mg Zn/kg DM from Zn_5_(OH)_8_Cl_2_·H_2_O. ^3^ SEM = standard error of the mean. ^4^ Mineral released post-dialysis = $\frac{\left(Initial\;concentration\;of\;digesta-Concentration\;of\;digesta\;post\;dialysis\right)}{Initial\;concentration\;of\;digesta}\times 100\%$. ^a,b,c,d,e^ Different superscripts between rows and columns within the element (Cu, Mn, Zn) indicate a significant difference between means; *p* < 0.05.

## Data Availability

The original contributions presented in the study are included in the article, further inquiries can be directed to the corresponding author.

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
