# Peer review of "Trace Mineral Source Impacts Volatile Fatty Acid Profile and Rumen Trace Mineral Solubility in Feedlot Steersâ€"

_animals, 2025, doi:10.3390/ani15091271_

Round 1
Reviewer 1 Report
Comments and Suggestions for Authors
The study is well designed and the manuscript is well written. Study can be of significant importance in the literature. The manuscript can be accepted after addressing some of the minor comments in the attached file.

Author Response
Dear reviewer, thank you for your review. Please see the attached for our responses to your comments.

Reviewer 2 Report
Comments and Suggestions for Authors
The manuscript is overall well written. One major strength of the manuscript is the industry relevance of the subject as it provides a validity for commercial application of mineral sources.
Only few minor comments are provided>
L137: VFA concentration
L138: How was ruminal content sampled?
L196/197: define the random variables.
Conclusion: Labelled sources of the mineral plus duodenal sampling would have provided a clue on the proportion of the minerals reaching the small intestine.
While solubility and performance is one context, digestion and absoption is also key as excretion footprint of minerals is becoming an important issue in sustainability.
Author Response

(The authors gave the same response as above.)

Reviewer 3 Report
Comments and Suggestions for Authors
Please find the attached review

Author Response
Thank you for taking the time to review this manuscript. Please find the detailed responses in the attached document and the corresponding revisions/corrections highlighted/in track changes in the re-submitted files.

Reviewer 4 Report
Comments and Suggestions for Authors
The study compares the supplementation of two Dietary treatments consisted of 18, 40, and 90 mg of supplemental Cu, Mn, and Zn/kg DM, respectively, in two presentations STM or HTM with a n of 6 steers/treatment, and they want to express Impacts Volatile Fatty Acid Molar Proportions and Rumen Trace Mineral Solubility in Steers, I do not see a control treatment to see how it affects or modifies the VFA profile in the rumen and its digestibility, why they think it would be affected.
Experiment 2 has higher concentrations of Cu, Mg and Mg in the rumen. The experiment 1 also has more concentrations of Cu, Mn, and Zn in the supplement were 620, 1350, 99 and 3020 mg/kg DM, respectively, with respect to the dose administered, so it is important to have a control diet without TM supplementation, and thus see its effect with TM and with two TM sources, in addition to the fact that it is an interaction of 3 minerals, the effect of their Ruminal soluble Cu concentrations were greater (p < 32 0.05) for STM-dosed steers. Steers receiving HTM had greater (p < 0.01) Cu and Zn and lesser (p < 33 0.01) Mn released from digesta compared to those receiving STM, was either Cu as such or due to the alteration, as there was no control, I consider that it loses validity, and and how do they separate effects, by Cu, Zn or Mn ? which mineral affects the fatty acid profile, Steers supplemented with HTM had greater (p < 0.05) ruminal molar proportions of propionate and lesser (p < 0.05) butyrate compared to STM supplemented steers ? This point should be discussed in depth, in my opinion it loses validity as there is no control to tell me what the VFA profile is like without the supplementation of TM, because more TM is supplemented with the corn ground `per se. ?
table 2 is incorrectly presented, because it is repeated measures with respect to time, in experiment 2. The correct analysis is repeated measures with respect to time, with the TM supplement and time and the animal as a covariate, however there is no control.
Author Response

(The authors gave the same response as above.)
